# Impact of Sarcopenia on the Survival of Patients with Hepatocellular Carcinoma Treated with Sorafenib

**DOI:** 10.3390/cancers16061080

**Published:** 2024-03-07

**Authors:** Maurizio Biselli, Nicola Reggidori, Massimo Iavarone, Matteo Renzulli, Lorenzo Lani, Alessandro Granito, Fabio Piscaglia, Stefania Lorenzini, Eleonora Alimenti, Giulio Vara, Paolo Caraceni, Angelo Sangiovanni, Massimo Marignani, Elia Gigante, Nicolò Brandi, Annagiulia Gramenzi, Franco Trevisani

**Affiliations:** 1Unit of Semeiotics, Liver and Alcohol-Related Diseases, IRCCS Azienda Ospedaliero-Universitaria di Bologna, 40124 Bologna, Italy; maurizio.biselli@unibo.it (M.B.); lorenzo.lani@studio.unibo.it (L.L.); stefania.lorenzini@aosp.bo.it (S.L.); paolo.caraceni@unibo.it (P.C.); annagiulia.gramenzi@unibo.it (A.G.); franco.trevisani@unibo.it (F.T.); 2Department of Medical and Surgical Sciences, University of Bologna, 40124 Bologna, Italy; alessandro.granito@unibo.it (A.G.); fabio.piscaglia@unibo.it (F.P.); 3Department of Internal Medicine, Ospedale per gli Infermi di Faenza, 48018 Faenza, Italy; 4Division of Gastroenterology and Hepatology, Foundation IRCCS Ca’ Granda Ospedale Maggiore Policlinico, 20122 Milan, Italy; massimo.iavarone@policlinico.mi.it (M.I.); eleonoralimenti@gmail.com (E.A.); angelo.sangiovanni@policlinico.mi.it (A.S.); 5Department of Radiology, IRCCS Azienda Ospedaliero-Universitaria di Bologna, 40138 Bologna, Italy; matteo.renzulli@unibo.it (M.R.); giulio.vara@studio.unibo.it (G.V.); nicolo.brandi@studio.unibo.it (N.B.); 6Division of Internal Medicine, Hepatobiliary and Immunoallergic Diseases, IRCCS Azienda Ospedaliero-Universitaria di Bologna, 40138 Bologna, Italy; 7Division of Gastroenterology and Hepatology, Ospedale Regina Apostolorum, 00041 Albano Laziale, Italy; mmarignani@hotmail.com; 8Service d’Hépato-Gastroentérologie et de Cancérologie Digestive, Hôpital Robert Debré, Université Reims-Champagne-Ardenne, 51092 Reims, France; egigante@chu-reims.fr

**Keywords:** hepatocellular carcinoma, sarcopenia, sorafenib, skeletal muscle index

## Abstract

**Simple Summary:**

Sarcopenia, conceived as low skeletal muscle mass and function, has been associated with worse outcomes in patients treated with Sorafenib for advanced HCC, with data coming mainly from the Oriental series. Skeletal muscle mass can be easily quantified from abdominal CT scans performed for advanced HCC staging. Sarcopenia and impaired liver function (MELD > 9) are strong predictors of unfavorable outcomes in patients affected by advanced HCC treated with Sorafenib. Their copresence can identify a subset of patients with particularly bad prognoses.

**Abstract:**

Background and aims: Sarcopenia has been associated with poor outcomes in patients with cirrhosis and hepatocellular carcinoma. We investigated the impact of sarcopenia on survival in patients with advanced hepatocellular carcinoma treated with Sorafenib. Methods: A total of 328 patients were retrospectively analyzed. All patients had an abdominal CT scan within 8 weeks prior to the start of treatment. Two cohorts of patients were analyzed: the “Training Group” (215 patients) and the “Validation Group” (113 patients). Sarcopenia was defined by reduced skeletal muscle index, calculated from an L3 section CT image. Results: Sarcopenia was present in 48% of the training group and 50% of the validation group. At multivariate analysis, sarcopenia (HR: 1.47, *p* = 0.026 in training; HR 1.99, *p* = 0.033 in validation) and MELD > 9 (HR: 1.37, *p* = 0.037 in training; HR 1.78, *p* = 0.035 in validation) emerged as independent prognostic factors in both groups. We assembled a prognostic indicator named “SARCO-MELD” based on the two independent prognostic factors, creating three groups: group 1 (0 prognostic factors), group 2 (1 factor) and group 3 (2 factors), the latter with significantly worse survival and shorter time receiving treatment.

## 1. Introduction

Hepatocellular carcinoma (HCC) is one of the most common causes of cancer-associated mortality worldwide, with a growing incidence on a global scale, around 8 cases per 100,000 people [1,2]. Unfortunately, only about 55% of HCCs are diagnosed at an early stage amenable to surgical or locoregional therapies [3,4]. Moreover, most early detected cancers progress to advanced stages for which the residual oncologic treatment relies on systemic therapies based on tyrosine-kinase inhibitors (TKI) or immunotherapy. Currently, immunotherapy is the treatment of choice but TKI still have a central role since a high proportion of patients are not eligible to receive immune checkpoint inhibitors [5]. In relation to this, Sorafenib still appears as a valid therapy and, to date, is the only drug that allows well-defined second- and third-line treatments outside of clinical trials [6]. Sorafenib exerts antitumoral effects by inhibiting RAF kinases in the Ras/Raf/MEK/ERK signaling pathway and the VEGF and PDGFR pathways [7]; furthermore, it induces autophagy in cancer cells [8]. In oncology, detection of reliable prognostic factors is a mainstay to define the correct treatment allocation for each patient. Defining the prognosis of HCC patients is challenging since it results from several determinants such as tumor burden, performance status (as assessed by Eastern Cooperative Oncology Group; ECOG), liver function (considering that most HCCs arise in the setting of liver cirrhosis), and amenability to curative treatments (liver transplant, hepatic resection, ablation) [1,9,10]. Several prognostic staging systems have been proposed for HCC, amongst which the updated version of the Barcelona Clinic Liver Cancer (BCLC) [6] is the most commonly used in Western countries for assessing prognosis and guiding treatment allocation.

Some studies have demonstrated that sarcopenia, defined as an age-related loss of skeletal muscle mass and function [11], is common and represents a pivotal prognostic factor for patients with liver cirrhosis and HCC [12,13]. Chronic liver disease and HCC are themselves important causes of secondary sarcopenia, promoting a catabolic state, affecting the endocrine balance, impairing the nutritional status, and promoting the synthesis of proinflammatory cytokines [14]. The negative impact of sarcopenia has been demonstrated in different therapeutic settings for HCC, such as liver transplantation [15,16], hepatic resection [17], and locoregional treatments [18,19]. Systemic therapy based on the TKI, Sorafenib has shown efficacy in improving survival in patients with advanced HCC and preserved liver function [20], although response and tolerability may vary notably between patients. Several studies on prognostic factors in patients with advanced HCC treated with Sorafenib focused attention on liver function, tumor features, and serum biomarkers [21,22,23]. Controversial data about the prognostic significance of sarcopenia in patients treated with Sorafenib are emerging from the literature, as most [24,25,26,27,28,29,30] but not all [31,32] studies show a negative impact of this variable on survival.

Moreover, data analyzing the role of sarcopenia as a predictor of reduced survival and tolerance to Sorafenib mainly come from Asian series.

Therefore, the aim of this study is to confirm that sarcopenia has an independent prognostic significance in European patients with HCC undergoing Sorafenib monotherapy and to validate the results on an independent HCC population. Furthermore, another objective is to define a predictive model of the death risk in this setting.

## 2. Materials and Methods

We retrospectively analyzed data prospectively collected for all consecutive patients managed by two Internal Medicine and Hepatology units of the IRCCS Azienda Ospedaliero-Universitaria of Bologna (i.e., Units of Semeiotics Liver and Alcohol-related diseases and Division of Internal Medicine, Hepatobiliary and Immunoallergic Diseases) between January 2007 and December 2021. These data were used as the training set. HCC diagnosis was made based on histology or non-invasive imaging criteria as suggested by the European Association for the Study of the Liver (1) and American Association for the Study of the Liver Disease (AASLD) guidelines [2] and staged by abdominal contrast-enhanced computed tomography (CT) scan or magnetic resonance imaging (MRI). Inclusion criteria were: (1) unresectable HCC at an advanced stage (BCLC-C); (2) unresectable HCC at an intermediate (BCLC-B) or early-stage (BCLC-A) with disease progression on locoregional therapies; (3) Sorafenib treatment as indicated by Agenzia Italiana del Farmaco (AIFA), namely Sorafenib was administrable only as a first-line monotherapy; (4) availability of an abdominal CT scan made within 8 weeks before Sorafenib initiation. For each patient, the following characteristics data were recorded at baseline (i.e., before Sorafenib initiation): age, gender, body mass index (BMI), presence of sarcopenia, etiology of cirrhosis, total bilirubin, serum albumin, international normalized ratio (INR), creatinine, platelet count, alpha-fetoprotein (AFP), presence of esophageal varices, ascites, and presence of metastases and/or macrovascular invasion (MVI). HCC was staged using BCLC, Cancer of the Liver Italian Program (CLIP) score [33], and the Prediction of Survival in Advanced Sorafenib-treated HCC-II (PROSASH-II) model [34]. Liver function was assessed according to Child-Turcotte-Pugh (CTP) score, Albumin-Bilirubin (ALBI) grade [35], and Mayo Model for End Stage Liver Disease (MELD) score. Performance status (PS) was calculated according to ECOG. Treatment duration and causes of its discontinuation were recorded. Adverse events were graded according to the National Cancer Institute’s Common Toxicity Criteria v4.0. Follow-up ended at the time of the patient’s death, at the last visit, or at the end of the study period. The need to obtain informed consent for inclusion in the study was waived because of the study’s retrospective design, not encompassing additional investigations or risks for patients. The study protocol was performed in accordance with the Declaration of Helsinki and with approval from the local Ethics Committee.

### 2.1. Image Analysis and Treatment Modality

Sarcopenia was defined as a reduction in the skeletal muscle index (SMI) assessed by a CT scan performed before Sorafenib initiation. As indicated [36], SMI was calculated on a CT transverse image crossing the 3rd lumbar vertebrae (L3). Skeletal muscles at this level included erector spinae, transverse abdominis, psoas, quadratus lumborum, external and internal oblique abdominal, and the rectus abdominis. Images were analyzed by an expert radiologist (MR) using ImageJ^®^ software (version 1.53s), which is a free software developed by the National Institute of Health (NIH). This software enables specific tissue selection by using previously determined Hounsfield unit (HU) intervals. Skeletal muscle is identified by thresholds between −29 to +150 HU. These specific thresholds allow for the identification of skeletal muscle regardless of ascites and visceral and subcutaneous adipose tissue in the selected section. The area occupied by skeletal muscle in the selected section is then normalized to the patient’s height, thus obtaining the SMI expressed as cm^2^/m^2^. According to previous studies conducted on a population with gastrointestinal and lung cancer [37], SMI was considered indicative of sarcopenia if <53 cm^2^/m^2^ for male patients with a BMI ≥ 25 and <43 cm^2^/m^2^ for those with a BMI < 25, and <41 cm^2^/m^2^ for women, regardless of BMI. Sorafenib was administered according to guidelines; dose was reduced, or treatment was temporarily withdrawn in the case of drug-related toxicity, while it was permanently discontinued in the case of cancer progression or unacceptable toxicity not reversible with dose reduction, as recommended by the manufacturer. Any concomitant therapy was prescribed according to general clinical practice. Local expert radiologists assessed the radiological response to treatment with a CT scan or MRI according to the modified RECIST criteria for solid tumors (mRECIST) [38].

### 2.2. Statistical Methods

The distribution asymmetry of quantitative data was assessed by skewness. Quantitative variables were expressed as mean ± standard deviation or median and interquartile range (IQR), as appropriate. Comparisons between groups were made by χ^2^ test or Fisher’s exact test (2-tailed) for qualitative variables, and the Mann–Whitney test or Kruskal–Wallis test for quantitative variables. The primary outcome was overall survival (OS). The Kaplan–Meier method was used to estimate survival curves which were compared using the log-rank test. Univariate and multivariate analyses for OS were made using the Cox proportional hazards model. Hazard ratio (HR), corresponding to a 95% confidence interval (95% CI) and the beta-coefficients with the bootstrap standard error were calculated for each variable. Variables associated with OS with *p* values < 0.1 in univariate analysis were entered into the multivariate analysis to identify the independent prognostic factors. A model of internal validation and adjustment for overfitting was implemented with bootstrapping procedures. To further reduce the risk of overfitting, we planned a model with at least five events (deaths) for each variable included [39]. Collinearity between variables was evaluated with variance inflation factor; to avoid redundancy, in case of collinear variables, only the most representative one was included in multivariate analysis (e.g., the individual components of the MELD score significantly associated with mortality at univariate analysis were not tested in the multivariate model including MELD score). If both CTP and MELD scores were significantly associated with OS at univariate analysis, only MELD was considered. The ability of each continuous variable to predict the mortality risk was assessed by the area under the receiver operating characteristic curve (AUROC) [40] and the cut-off values showing the best difference between true positive and false positive rates over all possible cut-point values [41] were used to dichotomize continuous variables. Concordance (c) statistics equivalent to the AUROC were used to assess the discrimination of a variable in predicting the death risk [40]. A two-tailed *p*-value < 0.05 was considered statistically significant. Statistical analysis was performed using SPSS-Windows 20.0 (IBM, SPSS-statistics) and STATA 12.0SE (StataCorp LP).

### 2.3. External Validation

The prognostic variables identified in the training set were validated in a cohort of cirrhotic patients with HCC followed by the Unit of Gastroenterology and Hepatology of the IRCCS Ca’ Granda Ospedale Maggiore, Policlinico, University of Milan, between January 2007 and April 2021. Inclusion and exclusion criteria, diagnostic criteria, and the indication of Sorafenib treatment were the same as were described for the training set. Methods to identify a significant association between variables and OS were the same as were used for the training set. Finally, the predictive performance of the prognostic variables was evaluated in the whole population (training plus validation set) of the study.

## 3. Results

### 3.1. Description of Patients and Drug Toxicity in Training and Validation Groups

Among 247 patients treated with Sorafenib for HCC in 2 Hepatology units of the IRCCS Azienda Ospedaliero-Universitaria of Bologna, 21 were excluded because a pre-treatment TC was not available, 10 because the CT scan available was performed beyond 8 weeks before treatment initiation, and 1 because the CT scan images were not examinable. The remaining 215 patients formed the training group.

Among 126 patients treated with Sorafenib at the “Ca’ Granda” Milan hospital, 7 were excluded because a pre-treatment TC was not available, and 10 because the CT scan available was performed beyond 8 weeks before treatment initiation. The remaining 113 patients served as a validation group. The baseline characteristics of all enrolled patients are described in Table 1. The two groups were not entirely matched, as differences were found in BMI, INR value, serum albumin, MELD score, AFP levels at baseline, and ECOG-PS. No differences were found in tumour burden except for a more frequent MVI in the training group (33.5% vs. 18.6%, *p* = 0.005). 

The mean SMI was not significantly different between the training and validation groups (47.7 ± 8.9 cm^2^/m^2^ and 45.8 ± 9.1 cm^2^/m^2^, respectively, *p* = 0.145), but showed significantly higher values in males than in females in both the training and validation group (48.8 ± 8.4 vs. 41.5 ± 9.2 cm^2^/m^2^, *p* < 0.001 and 48.5 ± 8 vs. 36.0 ± 5.6 cm^2^/m^2^, *p* < 0.001). 

Treatment duration did not significantly differ between the training and validation groups (135 [IQR 204] vs. 159 [IQR 199] days, respectively; *p* = 0.972). However, causes of Sorafenib discontinuation significantly differed (*p* = 0.044) with a preponderance of discontinuation for progressive disease in the training group (52.1% versus 39.8%, *p* = 0.037) and for unacceptable toxicity in the validation group (27.0% versus 40.7%, *p* = 0.013).

### 3.2. Description of Patients and Drug Toxicity according to the Presence/Absence of Sarcopenia

Table 2 describes the baseline characteristics of patients with or without sarcopenia in both the training and validation groups. Notably, the subdivision based on sarcopenia, led to significant differences as follows:in the training group, sarcopenic patients presented significantly more metastasis than non-sarcopenic ones (35.0% versus 19.7%; *p* = 0.014).in the validation group, sarcopenic patients showed lower male prevalence (66.1% versus 91.2%; *p* = 0.001), lower BMI (23.6 [IQR 5.48] versus 24.8 [IQR 4]; *p* = 0.043), and shorter Sorafenib treatment duration (103.5 [IQR 297.5] vs. 192 [IQR 297.5] days; *p* = 0.008).

Concerning the causes of Sorafenib discontinuation, no differences were observed according to sarcopenia in both the training and validation groups (Table 2).

Among patients that underwent Sorafenib dose reduction within the first month of therapy, no significant differences were observed based on the presence of sarcopenia in either group. Namely, of the 44 patients with a dose reduction in the training group, 26 (59.1%) were sarcopenic and 18 were not (*p* = 0.141); in the validation group, of 10 patients with a dose reduction, 5 were sarcopenic and 5 were not (*p* = 1).

### 3.3. Factors Associated with Overall Survival

#### 3.3.1. Training Group

The mean OS was 490 days (95% CI 409–570). At the end of the follow-up, 203 (94.4%) patients had died, 167 (82.3%) due to tumor progression, 17 (8.4%) due to liver failure, 8 (3.9%) due to variceal hemorrhage, 6 (3.0%) due to sepsis, and 5 (2.5%) due to non-liver-related causes. 

Using Cox univariate analysis, the following variables were selected as significantly associated with 540-day survival: bilirubin levels, presence of ascites, CTP score, MELD score, sarcopenia, ECOG-PS, AFP levels, presence of MVI, CLIP score, and PROSASH-II both as a continuous score and as categorized into 4 risk groups (Table 3A).

The best cut-offs for continuous variables significantly associated with OS in the univariate analysis were: AFP > 25 ng/mL (sensibility 60.4%, specificity 55.6%; *p* = 0.049) and MELD > 9 (sensibility 52.2%, specificity 67.9%; *p* = 0.009). Cox univariate analysis confirmed the prognostic significance of the two variables even when they were dichotomized using these cut-offs (Table 3A).

Variables selected by the multivariate Cox model as independent predictors of 540 days survival were: the presence of sarcopenia and MELD score > 9 (Table 3A).

Survival rates according to the presence of sarcopenia and MELD score were evaluated; namely, the 270- and 540-day survival rates of patients with sarcopenia (51.5% and 24%, respectively) were lower than those of non-sarcopenics (61.6% and 39.9%, respectively; *p* = 0.023; Figure 1(aT)). Similarly, 270- and 540-day survival rates of patients with MELD score > 9 (51% and 25%, respectively) were lower than those of patients with MELD score ≤ 9 (61.3% and 38.1%, respectively; *p* = 0.014; Figure 1(bT)).

#### 3.3.2. Validation Group

The mean OS in the validation group was 459 days (95% CI 371–548). At the end of the follow-up, all patients were dead. The leading cause of death was tumor progression (103 patients, 91.2%) followed by liver failure (9 patients, 8%), while only 1 patient (0.9%) died from a non-liver-related cause. 

Using Cox univariate analysis, the following variables were significantly associated with 540-day survival: age, albumin levels, presence of sarcopenia, CTP score, MELD score, AFP levels, PROSASH-II groups, and ECOG-PS (Table 3B). AFP levels and MELD scores were dichotomized according to cut-offs identified in the training group, whereas for age and albumin, the best cut-off values were >75 years (sensibility 79.0%, specificity 40.6%; *p* = 0.167) and <3.8 g/dL (sensibility 61.7%, specificity 62.5%; *p* = 0.136), respectively. These categorized variables were tested in the univariate analysis and were shown to be prognostic predictors (Table 3B).

Variables selected in the multivariate Cox model as independent predictors of survival at 540 days were: presence of sarcopenia, MELD score > 9, and ECOG-PS (Table 3B). Therefore, the only variables showing an independent prognostic meaning both in the training and validation groups were sarcopenia and a MELD score > 9.

The 270- and 540-day survival rates of patients with sarcopenia (50% and 19.6%, respectively) were lower than those of patients without sarcopenia (71.9% and 36.8% respectively; *p* = 0.011; Figure 1(aV)). Similarly, 270- and 540-day survival rates of patients with a MELD score > 9 (40% and 17.1%, respectively) were lower than those of patients with a MELD score ≤ 9 (70.5% and 33.3%, respectively; *p* = 0.003; Figure 1(bV)).

#### 3.3.3. SARCO-MELD Prognostic Model

We assembled an easy-to-use prognostic indicator named “SARCO-MELD” based on the two independent risk factors identified in both the training and validation groups (MELD > 9 and presence of sarcopenia). According to this model, three SARCO-MELD grades with a different risk of death were created:SARCO-MELD grade 1 (low-risk group, 95 patients): characterized by the absence of both risk factors.SARCO-MELD grade 2 (intermediate-risk group, 158 patients): characterized by the presence of 1 of the 2 risk factors.SARCO-MELD grade 3 (high-risk group, 58 patients): characterized by the presence of both risk factors.

After including the SARCO-MELD grade, instead of MELD > 9 and sarcopenia, in the Cox multivariate analysis, this new composite variable became the only variable independently associated with 540 days survival in both the training and validation groups (Table 4).

Pooling training and validation groups, we observed a progressive decline of median OS across SARCO-MELD grades (grade 1: median OS 482 days, 95% CI 354–610 days; grade 2: median OS 308 days, 95% CI 275–341 days; grade 3: median OS 201 days, 95% CI 152–250 days). This decline was statistically significant when grade 1 patients were compared to grade 2 (*p* = 0.002) and grade 3 (*p* < 0.001) patients, as well as when grade 2 and 3 patients were compared (*p* = 0.003) (Figure 2).

The corresponding 270- and 540-day survival rates were 70.7% and 45.0%, respectively, for grade 1 patients, 58.3% and 28.4% for grade 2 patients, and 37.7% and 14.8% for grade 3 patients.

Moreover, the three SARCO-MELD grades showed a significant difference in treatment duration: namely, a progressive decline of treatment duration occurred across SARCO-MELD grades (grade 1: 192 days, IQR 266 days; grade 2: 121 days, IQR 202 days; grade 3: 71 days, IQR 143 days; *p* < 0.001). Conversely, no significant difference was observed in the presence of adverse events among the three risk groups (87.9%, 86.9%, and 77%, respectively; *p* = 0.123).

Finally, SMI thresholds associated with low survival were: ≤53.9 cm^2^/m^2^ for overweight or obese male patients; ≤42.2 cm^2^/m^2^ for those of normal weight or under-weight, and ≤40.8 cm^2^/m^2^ for females regardless of BMI.

## 4. Discussion

Sarcopenia is a common feature in patients with cirrhosis [12] and HCC [42]; in these circumstances, Sorafenib may further increase skeletal muscle depletion through the inhibition of Ras-dependent muscle differentiation [43], the impairment of carnitine absorption [44], and by enhancing autophagy [45]. A rising amount of literature highlights the negative impact of muscle mass depletion in these populations.

Focusing on HCC patients undergoing systemic therapies, the available studies provide controversial results, as some describe a negative impact of baseline sarcopenia whereas others do not. Moreover, to our knowledge, only two small-sized studies have analyzed the influence of sarcopenia on Sorafenib tolerability and efficacy in Caucasian patients [28,46], reporting a lower tolerance rate and a negative impact on survival.

As far as we know, this is the largest multicentric study evaluating the impact of sarcopenia on the survival of European patients treated with Sorafenib for HCC and is the only one where data were externally validated in an independent cohort of patients (validation group). Of note, baseline characteristics were not completely matched in the two cohorts (training and validation groups) with significant differences in terms of liver function, tumor burden, and causes of Sorafenib discontinuation. These diversities could be interpreted as a strong point of our results since the prognostic factors identified maintain their validity regardless of the baseline characteristics of HCC patients.

In line with previous studies [25,28,31], the prevalence of sarcopenia, conceived as an SMI value below a validated threshold, was high in HCC patients amenable to Sorafenib therapy, being around 50% in both the training and validation groups. This high prevalence was not surprising since our patients were affected by two of the main causes of secondary sarcopenia, i.e., cirrhosis and cancer.

Of note, in the training set, treatment duration did not differ between sarcopenic and non-sarcopenic patients, whereas in the validation cohort, treatment duration was significantly shorter in sarcopenic patients. We do not have a clear explanation for this difference, which might be attributed to an imbalance between baseline patient features and causes of treatment discontinuation between the two centres, namely a greater prevalence of cancer progression in the training set and of toxicity in the validation set.

The variables with prognostic significance both in the training and validation groups were sarcopenia and MELD score > 9, confirming the importance of nutritional status and residual liver function in determining the prognosis of Sorafenib-treated HCCs, as already reported [24,28]. Concerning the evaluation of residual liver function, we chose to utilize MELD instead of CTP score because of its limits due to the “ceiling effect” of laboratory parameters and the subjective assessment of ascites and hepatic encephalopathy.

Some studies conducted in cirrhotic patients suggest the beneficial prognostic effect of physical exercise [47] and nutritional supplementation with branched-chain amino acids [48]. Therefore, considering the high frequency and the negative impact of sarcopenia in HCC patients undergoing systemic therapy, studies analyzing the effect of physical exercise and nutritional supplementation in this setting are warranted.

A key result of our study was the creation of a simple and easy-to-obtain prognostic model, named SARCO-MELD, which was able to allocate patients treated with Sorafenib to three prognostic groups with significantly different survival probabilities (Figure 2). Notably, SARCO-MELD grade 3 (presence of sarcopenia and MELD score > 9) identifies a subset of patients with a very short Sorafenib treatment duration and a low survival probability, like that of placebo-treated patients of the SHARP trial [20]. Considering the short treatment duration observed in SARCO-MELD grade 3 patients, a survival benefit from Sorafenib is unlikely to be obtained in this subgroup, and this study brings evidence against the use of Sorafenib in these patients.

Lastly, although the correlation between muscle depletion and survival has already been reported in European [28,46,49] and Asian patients with HCC [25,26,29,31], the cut-off values of SMI predicting poor outcome vary among regions and ethnicities. Since no consensus for CT-based SMI values suggestive of sarcopenia has been established for the European population, we used the cut-offs defined in a study conducted on a large cohort of Caucasian patients affected by gastrointestinal or lung cancer with a wide BMI distribution, like our population. Intriguingly, the SMI thresholds prognostically subdividing our population at best were highly comparable with those proposed by the Martin et al. study [37], confirming the validity and reproducibility of the previously identified cut-offs.

There are several limitations to this study. This was a retrospective study and consequently, the impact of unintended biases and confounding factors cannot be excluded. Moreover, the retrospective design kept us from evaluating whether sarcopenia was an independent prognostic factor even in patients treated with only supportive care, and therefore we cannot give a stronger indication against the use of Sorafenib in patients with sarcopenia and a MELD score > 9. In addition, although the management of HCC patients on Sorafenib followed the European and Italian guidelines in the participating centres, it was not pre-established and, therefore, some inter-centre differences cannot be excluded. In addition, data about the food intake of patients were not available; therefore, the impact of diet on outcomes was not able to be explored. Finally, in this study, data regarding muscle mass were collected by retrospective analysis of CT scan and the inter-observer reproducibility of the SMI evaluation was not tested. However, since SMI is automatically calculated by the ImageJ^®^ software (version 1.53s), this methodology allows excellent intra-observer reproducibility.

## 5. Conclusions

In conclusion, this study demonstrated that sarcopenia is an independent prognostic factor in patients with HCC undergoing Sorafenib therapy, and its association with a MELD score > 9 identifies patients with a very poor outcome despite the treatment.

## Figures and Tables

**Figure 1 cancers-16-01080-f001:**
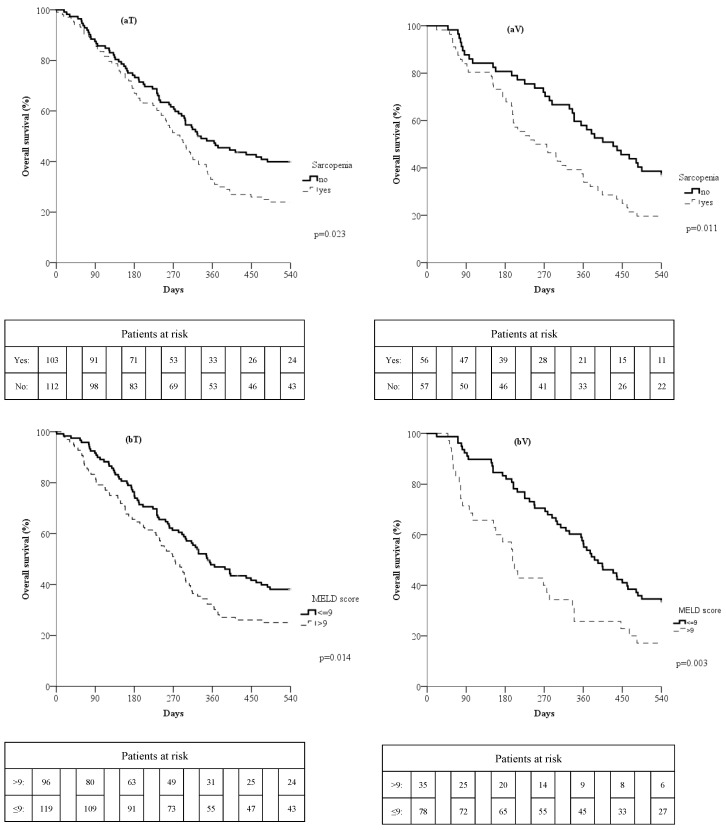
Overall survival curves in the training (**T**) and validation (**V**) groups according to the presence of sarcopenia (**a**) or MELD > 9 (**b**).

**Figure 2 cancers-16-01080-f002:**
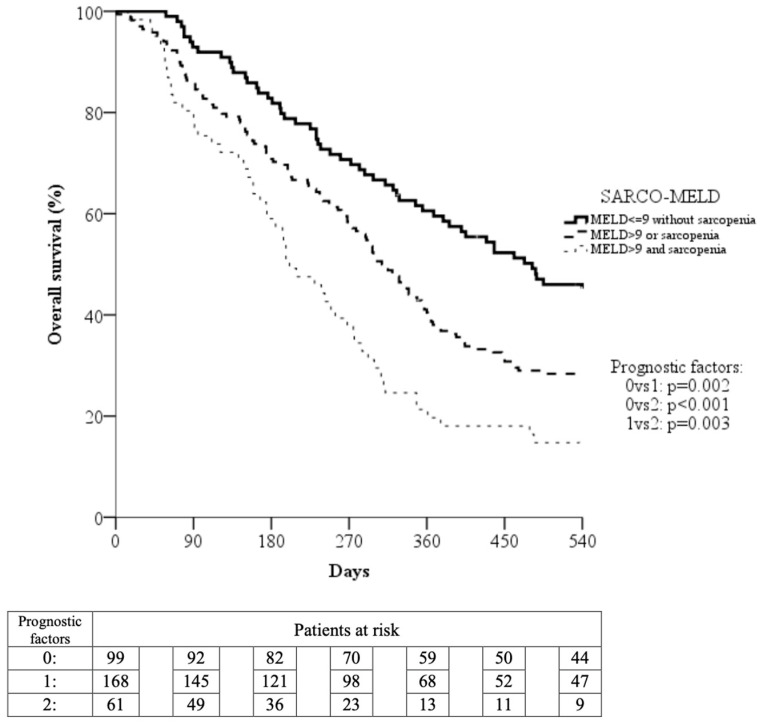
Overall survival curves evaluated on the entire cohort of study patients subdivided based on presence/absence of the two prognostic factors: MELD > 9 and sarcopenia.

**Table 1 cancers-16-01080-t001:** Patients’ baseline characteristics in the training and validation groups.

	Training Group (n = 215)	Validation Group (n = 113)	*p*-Value
Age (years)	69 (IQR 14)	68 (IQR 13,5)	0.912
Gender (M/F)	183/32 (85.1%/14.9%)	89/24 (78.8%/21.2%)	0.165
Sarcopenia (yes/no)	103/112 (47.9%/52.1%)	56/57 (49.6%/50.4%)	0.817
Etiology of cirrhosis - HCV - HBV - Alcohol - Others			0.077
	85 (39.5%)	54 (47.8%)	
	34 (15.8%)	17 (15.0%)	
	27 (12.6%)	20 (17.7%)	
	69 (32.1%)	22 (19.5%)	
BMI	25.62 (IQR 5.7)	24.60 (IQR 4.6)	**0.001**
Bilirubin (mg/dL)	1.02 (IQR 0.8)	0.97 (IQR 0.83)	0.695
INR	1.16 (IQR 0.18)	1.00 (IQR 0.16)	**<0.001**
Albumin (g/dL)	3.5 (IQR 0.61)	3.8 (IQR 0.6)	**0.036**
Creatinine (mg/dL)	0.85 (IQR 0.32)	0.80 (IQR 0.30)	0.557
Esophageal varices	107 (49.8%)	65 (57.5%)	0.201
Platelets (×10^9^/mmc)	129 (IQR 107)	123 (IQR 122)	0.406
Ascites	63 (29.3%)	23 (20.4%)	0.087
ALBI score	−2.19 (IQR 0.69)	−2.4 (IQR 0.60)	0.051
MELD score	9 (IQR 3)	8 (IQR 4)	**0.002**
CTP class - A - B			0.214
	161 (74.9%)	92 (81.4%)	
	54 (25.1%)	21 (18.6%)	
AFP baseline	36 (IQR 538.2)	45 (IQR 487.0)	**0.045**
Macrovascular invasion	72 (33.5%)	21 (18.6%)	**0.005**
Metastasis	58 (27%)	26 (23%)	0.506
ECOG PS - 0 - 1 - 2			**0.013**
	108 (50.2%)	40 (35.4%)	
	78 (36.3%)	46 (40.7%)	
	29 (13.5%)	27 (23.9%)	
BCLC stage - A - B - C			0.340
	3 (1.4%)	0	
	36 (16.7%)	23 (20.4%)	
	176 (81.9%)	90 (79.6%)	
CLIP score - 0 - 1 - 2 - 3 - 4 - 5			0.706
	22 (10.2%)	11 (9.7%)	
	60 (27.9%)	25 (22.1%)	
	74 (34.4%)	41 (36.3%)	
	44 (20.5%)	30 (26.5%)	
	14 (6.5%)	6 (5.3%)	
	1 (0.5%)	0	
PROSASH-II model	0.325 ± 0.459	0.257 ± 0.4	0.211
PROSASH-II risk groups - 1 - 2 - 3 - 4			0.443
	39 (18.1%)	22 (19.5%)	
	79 (36.7%)	49 (43.4%)	
	67 (31.2%)	32 (28.3%)	
	30 (14%)	10 (8.8%)	
Treatment duration (days)	135.0 (IQR 204.0)	159 (IQR 199.0)	0.972
Presence of adverse effects during Sorafenib - Severe	184 (85.6%)	96 (85.0%)	0.871
	59 (27.4%)	36 (31.9%)	0.443
Cause of Sorafenib discontinuation - No suspension - Progressive disease - Adverse effects - Independent cause - Death			**0.044**
	5 (2.3%)	0	
	112 (52.1%)	45 (39.8%)	
	58 (27.0%)	46 (40.7%)	
	15 (7.0%)	10 (8.8%)	
	25 (11.6%)	12 (10.6%)	

Data are expressed as medians and interquartile ranges (IQRs) (in parenthesis) or percentages. Comparisons between groups were made by χ^2^ test or Fisher’s exact test (2-tailed) for qualitative variables, and the Mann–Whitney test or Kruskal–Wallis test for quantitative variables. Significant *p*-values are highlighted in bold. Abbreviations: AFP, Alpha-fetoprotein; ALBI, Albumin-Bilirubin; BCLC, Barcelona Clinic Liver Cancer; BMI, Body Mass Index; CLIP, Cancer Liver Italian Program; PROSASH-II, Prediction of Survival in Advanced Sorafenib-treated HCC-II; CTP, Child-Turcotte-Pugh; ECOG-PS, Eastern Cooperative Oncology Group Performance Status; INR, International Normalized Ratio; MELD, Model for End-stage Liver Disease.

**Table 2 cancers-16-01080-t002:** Patients’ baseline characteristics in the training and validation groups subdivided by presence of sarcopenia.

	Training Group (n = 215)			Validation Group (n = 113)		
	Sarcopenic Group (n = 103)	Non-Sarcopenic Group (n = 112)	*p*	Sarcopenic Group (n = 56)	Non-Sarcopenic Group (n = 57)	*p*
Age	69 (IQR 13.0)	68 (IQR 16.8)	0.211	71.5 (IQR 13.0)	67.0 (IQR 16.0)	0.073
Sex (male)	87 (84.5%)	96 (85.7%)	0.849	37 (66.1%)	52 (91.2%)	**0.001**
Etiology - HCV - HBV - Alcohol - Other			0.176			**0.018**
	46 (44.7%)	39 (34.8%)		33 (58.9%)	21 (36.8%)	
	19 (18.4%)	15 (13.4%)		9 (16.1%)	8 (14.0%)	
	10 (9.7%)	17 (15.2%)		4 (7.1%)	16 (28.1%)	
	28 (27.2%)	41 (36.6%)		10 (17.9%)	12 (21.1%)	
BMI	26.06 (IQR 5.65)	24.9 (IQR 6.13)	0.699	23.55 (IQR 5.48)	24.8 (IQR 4.00)	**0.043**
INR	1.15 (IQR 0.18)	1.16 (IQR 0.20)	0.482	1.10 (IQR 0.17)	1.08 (IQR 0.17)	0.984
Ascites	35 (34%)	28 (25%)	0.177	12 (21.4%)	11 (19.3%)	0.819
Bilirubin (mg/dL)	1.02 (IQR 0.84)	1.04 (0.83)	0.282	1.00 (IQR 0.85)	0.90 (IQR 0.80)	0.640
Albumin (g/dL)	3.5 (IQR 0.60)	3.5 (IQR 0.68)	0.280	3.80 (IQR 0.78)	3.80 (IQR 0.70)	0.175
ALBI score	−2.16 (IQR 0.73)	−2.22 (IQR 0.68)	0.626	−2.27 (0.79)	−2.40 (0.58)	0.119
CTP class - A - B			0.348			0.235
	74 (71.8%)	87 (77.7%)		43 (76.8%)	49 (86.0%)	
	29 (28.2%)	25 (22.3%)		13 (23.2%)	8 (14.0%)	
MELD	9 (IQR 3)	9 (IQR 3)	0.349	8 (IQR 3)	8 (IQR 3)	0.542
Presence of varices	57 (55.3%)	50 (44.6%)	0.134	34 (60.7%)	31 (54.4%)	0.570
Platelets (×10^9^/mmc)	138 (IQR 109)	123.5 (IQR 96.5)	0.153	126.5 (IQR 95.5)	113 (IQR 146)	0.274
ECOG-PS - 0 - 1 - 2			0.693			0.192
	50 (48.5%)	58 (51.8%)		19 (33.9%)	21 (36.8%)	
	37 (35.9%)	41 (36.6%)		27 (48.2%)	19 (33.3%)	
	16 (15.5%)	13 (11.6%)		10 (17.9%)	17 (29.8%)	
AFP (ng/mL)	43.0 (IQR 484.0)	30.4 (IQR 549.5)	0.825	67 (IQR 547.8)	58 (IQR 465.5)	0.703
Macrovascular invasion	39 (37.9%)	33 (29.5%)	0.197	12 (21.4%)	9 (15.8%)	0.477
Metastasis	36 (35.0%)	22 (19.6%)	**0.014**	14 (25.0%)	12 (21.1%)	0.660
BCLC - A - B - C			0.066			0.819
	3 (2.9%)	0				
	13 (12.6%)	23 (20.5%)		12 (21.4%)	11 (19.3%)	
	87 (84.5%)	89 (79.5%)		44 (78.6%)	46 (80.7%)	
CLIP - 0 - 1 - 2 - 3 - 4 - 5			0.479			0.271
	13 (12.6%)	9 (8.0%)		4 (7.1%)	7 (12.3%)	
	28 (27.2%)	32 (28.6%)		14 (25.0%)	11 (19.3%)	
	35 (34.0%)	39 (34.8%)		17 (30.4%)	24 (42.1%)	
	23 (22.3%)	21 (18.8%)		19 (33.9%)	11 (19.3%)	
	4 (3.9%)	10 (8.9%)		2 (3.6%)	4 (7.0%)	
	0	1 (0.9%)		0	0	
PROSASH-II model	0.341 ± 0.484	0.311 ± 0.436	0.565	0.286 ± 0.392	0.227 ± 0.41	0.312
Treatment duration (days)	116 (IQR 196)	142 (IQR 203.3)	0.152	103.5 (IQR 297.5)	192.0 (IQR 297.5)	**0.008**
Adverse effects (AE)	84 (81.6%)	100 (89.3%)	0.122	47 (83.9%)	49 (86.0%)	0.798
Severe AE	31 (30.1%)	28 (25.0%)	0.446	19 (33.9%)	17 (29.8%)	0.689
Cause of Sorafenib discontinuation - Still on treatment - Progressive disease - Adverse effects - Independent cause - Death			0.078			0.645
	2 (1.9%)	3 (2.7%)		0	0	
	46 (44.7%)	66 (58.9%)		20 (35.7%)	25 (43.9%)	
	28 (27.2%)	30 (26.8%)		26 (46.4%)	20 (35.1%)	
	10 (9.7%)	5 (4.5%)		5 (8.9%)	5 (8.8%)	
	17 (16.5%)	8 (7.1%)		5 (8.9%)	7 (12.3%)	

Data are expressed as medians and interquartile ranges (IQRs) (in parenthesis) or percentages. Comparisons between groups were made by χ^2^ test or Fisher’s exact test (2-tailed) for qualitative variables, and the Mann-Whitney test or Kruskal–Wallis test for quantitative variables. Significant *p*-values are highlighted in bold. Abbreviations: AFP, Alpha-fetoprotein; ALBI, Albumin-Bilirubin; BCLC, Barcelona Clinic Liver Cancer; BMI, Body Mass Index; CLIP, Cancer Liver Italian Program; CTP, Child-Turcotte-Pugh; ECOG-PS, Eastern Cooperative Oncology Group Performance Status; INR, International Normalized Ratio; MELD, Model for End-stage Liver Disease; PROSASH-II, Prediction of Survival in Advanced Sorafenib-treated HCC-II.

**Table 3 cancers-16-01080-t003:** Cox multivariate analysis for prognostic factors for survival 18 months after Sorafenib initiation. (A) training group. (B) validation group.

**(A) Training Group n = 215**				
	**Univariate Analysis**		**Multivariate Analysis**	
**Variable**	**HR (95% CI)**	***p*-Value**	**HR (95% CI)**	***p*-Value**
Gender (Male)	1.05 (0.68–1.60)	0.838		
Age (yr)	0.99 (0.97–1.01)	0.456		
BMI	0.99 (0.96–1.03)	0.785		
INR	1.03 (0.35–3.00)	0.963		
Bilirubin (mg/dL)	1.35 (1.14–1.59)	**<0.001**		
Albumin (g/dL)	0.80 (0.54–1.17)	0.242		
Creatinine (mg/dL)	1.02 (0.75–1.39)	0.890		
Platelets (×10^9^)	1.00 (0.99–1.00)	0.965		
Ascites	1.63 (1.15–2.33)	**0.006**	1.35 (0.93–1.97)	0.119
Oesophageal varices	1.08 (0.79–1.49)	0.625		
CTP class	1.39 (1.14–1.68)	**0.001**		
MELD	1.08 (1.02–1.16)	**0.011**		
MELD > 9	1.50 (1.04–2.17)	**0.030**	1.37 (1.02–1.83)	**0.037**
ALBI score	1.12 (0.70–1.79)	0.633		
Sarcopenia	1.45 (1.10–1.93)	**0.008**	1.47 (1.05–2.07)	**0.026**
ECOG–PS	1.32 (1.05–1.65)	**0.017**	1.18 (0.93–1.51)	0.178
AFP	1.000019 (1.000014–1.000024)	**<0.001**		
AFP (>25 ng/mL)	1.72 (1.25–2.38)	**0.001**	1.42 (0.97–2.07)	0.07
Macrovascular invasion	1.70 (1.26–2.30)	**0.001**	1.29 (0.85–1.95)	0.228
Metastasis	1.07 (0.71–1.60)	0.762		
BCLC	1.35 (0.90–2.04)	0.150		
PROSASH-II	2.52 (1.62–3.93)	**<0.001**		
PROSASH-II risk groups	1.47 (1.2–1.8)	**<0.001**	1.2 (0.95–1.52)	0.134
CLIP	1.58 (1.18–2.13)	**0.002**	1.07 (0.87–1.3)	0.521
**(B) Validation Group n = 113**				
	**Univariate Analysis**		**Multivariate Analysis**	
**Variable**	**HR (95% CI)**	** *p* ** **-Value**	**HR (95% CI)**	** *p* ** **-Value**
Gender (Male)	1.09 (0.66–1.79)	0.744		
Age (yr)	0.97 (0.95–1)	**0.041**		
Age > 75 yr	0.58 (0.33–1.01)	**0.055**	0.57 (0.29–1.15)	0.115
BMI	1.00 (0.95–1.06)	0.888		
INR	4.26 (0.71–25.59)	0.113		
Bilirubin (mg/dL)	1.30 (0.91–1.87)	0.155		
Albumin (g/dL)	0.58 (0.36–0.94)	**0.028**		
Albumin < 3.8 g/dL	1.80 (1.25–2.58)	**0.002**	1.58 (0.79–3.18)	0.2
Creatinine (mg/dL)	2.07 (0.65–6.59)	0.217		
Platelets (×10^9^)	1.00 (0.99–1.01)	0.190		
Ascites	1.01 (0.60–1.71)	0.965		
Oesophageal varices	0.85 (0.59–1.23)	0.380		
CTP class	1.37 (0.98–1.90)	**0.064**		
MELD > 9	1.96 (1.14–3.38)	**0.015**	1.78 (1.04–3.03)	**0.035**
ALBI score	1.38 (0.86–2.22)	0.182		
Sarcopenia	1.75 (1.17–2.63)	**0.007**	1.99 (1.06–3.7)	**0.033**
ECOG–PS	1.50 (1.17–1.93)	**0.001**	1.51 (1.05–2.18)	**0.028**
AFP > 25 ng/ml	1.75 (1.04–2.87)	**0.035**	1.1 (0.6–2)	0.765
Macrovascular invasion	1.07 (0.59–1.94)	0.819		
Metastasis	0.59 (0.28–1.26)	0.174		
BCLC	1.37 (0.67–2.82)	0.393		
PROSASH-II risk groups	1.27 (0.99–1.62)	**0.063**	1.04 (0.68–1.59)	0.852
CLIP	1.30 (0.78–2.17)	0.312		

Data are expressed as Hazard Ratio and 95% Confidence Interval. Univariate and multivariate analyses for OS were made using the Cox proportional hazards model. Significant *p*-values are highlighted in bold. AFP, Alpha-fetoprotein; ALBI, Albumin-Bilirubin; BCLC, Barcelona Clinic Liver Cancer; BMI, Body Mass Index; CLIP, Cancer Liver Italian Program; CTP, Child-Turcotte-Pugh; ECOG-PS, Eastern Cooperative Oncology Group Performance Status; INR, International Normalized Ratio; MELD, Model for End-stage Liver Disease; PROSASH-II, Prediction of Survival in Advanced Sorafenib-treated HCC-II.

**Table 4 cancers-16-01080-t004:** Cox multivariate analysis of factors associated with poor survival 18 months after Sorafenib initiation using SarcoMELD composite variable. (A) Training group; (B) Validation group.

**(A) Training Group**		
**Variable**	**HR (95% CI)**	***p*-Value**
SarcoMELD	1.42 (1.12–1.8)	**0.004**
AFP > 25 ng/mL	1.43 (0.99–2.05)	0.053
Ascites	1.35 (0.89–2.03)	0.157
Macrovascular invasion	1.3 (0.86–1.98)	0.213
ECOG–PS	1.19 (0.95–1.47)	0.124
CLIP	1.06 (0.85–1.33)	0.583
PROSASH-II risk groups	1.89 (0.89–1.58)	0.237
**(B) Validation Group**		
**Variable**	**HR (95% CI)**	** *p* ** **-Value**
SarcoMELD	1.88 (1.17–3.04)	**0.009**
AFP > 25 ng/mL	1.09 (0.69–1.73)	0.696
Albumin < 3.8 g/dL	1.59 (0.86–2.97)	0.142
Age > 75 years	0.58 (0.3–1.11)	0.1
ECOG–PS	1.5 (1.12–2.01)	**0.006**
PROSASH-II risk groups	1.03 (0.72–1.48)	0.875

Data are expressed as Hazard Ratio and 95% Confidence Interval. Multivariate analysis for OS was made using the Cox proportional hazards model. Significant *p*-values are highlighted in bold. Abbreviations: AFP, Alpha-fetoprotein; CLIP, Cancer Liver Italian Program; ECOG-PS, Eastern Cooperative Oncology Group Performance Status; MELD, Model for End stage Liver Disease; PROSASH-II, Prediction of Survival in Advanced Sorafenib-treated HCC-II.

## Data Availability

All data generated or analyzed during this study are included in this article. Further inquiries can be directed to the corresponding author.

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
