# Peer review of "Impact of Sarcopenia on the Survival of Patients with Hepatocellular Carcinoma Treated with Sorafenib"

_cancers, 2024, doi:10.3390/cancers16061080_

Round 1

Reviewer 1 Report

Comments and Suggestions for Authors

The authors investigated the impact of sarcopenia on survival in patients with advanced hepatocellular carcinoma treated with sorafenib.

In the initial part of the introduction section, please add incidence of hepatocellular carcinoma worldwide.

The authors could mention the first-line therapy for patients with advanced hepatocellular carcinoma and its drawbacks if any.

"Sorafenib suppresses tumor cell proliferation by inhibiting Raf-1, B-Raf, and kinase activity in the Ras/Raf/MEK/ERK signaling pathways."  These facts need to be discussed.

Is Sorafenib more effective in the late stages of HCC?

It is important to discuss the effect of Sorafenib on serum levels of carnitine since we are dealing with sarcopenia.

 What are the chances of patients developing resistance? How to tackle such conditions of resistance?

 Does Sorafenib interact with other drugs?  What about interaction with ABC transporters?

 Autophagy in sorafenib resistance can be discussed a bit.

Can combinations of checkpoint inhibitors and other drugs be used for advanced HCC?

What are the evidences for dose-dependent antitumour activity?

Sarcopenia predicts the occurrence of dose limiting toxicities within the first month of sorafenib therapy. The authors need to discuss such.

 Did the authors look into food intake, anorexia as these are common complications?

Comments on the Quality of English Language

Minor errors should be rectified.

Author Response

The authors investigated the impact of sarcopenia on survival in patients with advanced hepatocellular carcinoma treated with sorafenib.

  1. In the initial part of the introduction section, please add incidence of hepatocellular carcinoma worldwide.

    Author Response: as suggested by the reviewer we have added the incidence of hepatocellular carcinoma in the Introduction section of the amended version of the manuscript (page 2, lines 50-51)
  2. The authors could mention the first-line therapy for patients with advanced hepatocellular carcinoma and its drawbacks if any.

    Author Response: following the suggestion of the reviewer we have described the landscape of first-line therapy for patients with advanced hepatocellular carcinoma in the Introduction section of the amended version of the manuscript (page 2, lines 54-58)

  3. "Sorafenib suppresses tumor cell proliferation by inhibiting Raf-1, B-Raf, and kinase activity in the Ras/Raf/MEK/ERK signaling pathways."  These facts need to be discussed. It is important to discuss the effect of Sorafenib on serum levels of carnitine since we are dealing with sarcopenia. Is Sorafenib more effective in the late stages of HCC?

    Author Response: We agree with the reviewer’s comment about sorafenib mechanism of action. Accordingly, throughout the manuscript, we have described in more depth detail the effects of sorafenib on Ras/Raf/MEK/ERK signaling pathways (page 2, lines 58-60; page 5, lines 105-108) and carnitine serum levels (page 13, line 347). 
    Unfortunately, whether sorafenib shows a better efficacy in the late stage of HCC could not be evaluated in this study, because it was not designed to demonstrate this hypothesis. However, we report in this comment the data we can provide, even not exhaustive, for the question: survival in patients with HCC staged in BCLC grade C was slightly inferior compared to patients with HCC staged in BCLC grade A and B, even if this difference did not reach a statistical significance (mean survival 315 days, 95%CI 293-337, vs 370 days, 95%CI 329-411; respectively; p=0.087) (data not shown in the paper).

  4. What are the chances of patients developing resistance? How to tackle such conditions of resistance? Does Sorafenib interact with other drugs? What about interaction with ABC transporters?

    Author Response: We appreciate the suggestions of the reviewer and we agree that it would be interesting to discuss sorafenib resistance as well as mechanisms of resistance and drug-drug interactions, however we believe that such aspects are beyond the scope of our paper, which aims only to show that sarcopenia has a negative effect on patients treated with sorafenib

  5. Autophagy in sorafenib resistance can be discussed a bit.

    Author Response: Done as requested (page 2, line 60). The possible conection between induction of autophagy an worsening of sarcopenia has also been discussed (page 13, line 347). We thank the reviewer for the hint. 

  6. Can combinations of checkpoint inhibitors and other drugs be used for advanced HCC? What are the evidences for dose-dependent antitumour activity?

    Author Response: although we are grateful for your proposal to incorporate more data about therapy of advanced hepatocellular carcinoma and dose-dependent antitumor activity, we respectfully disagree that they are required given the nature of our research as previously explained (see point 4).

  7. Sarcopenia predicts the occurrence of dose limiting toxicities within the first month of sorafenib therapy. The authors need to discuss such.

    Author Response: thank you for pointing this out. As suggested, in the amended version of the manuscript we have added and discussed data on the occurrence of dose limiting toxicities within the first month of sorafenib therapy (page 6, lines 227-231). The data didn't show a clear correlation between sarcopenia and dose limiting toxicities in the first month, probably in correlation with the small sample size.

  8. Did the authors look into food intake, anorexia as these are common complications?

    Author response: Thank you for this suggestion. It would have been interesting to explore this aspect. Unfortunately, we do not have the required data also because of the retrospective nature of the study.  We agree that this is a potential limitation and accordingly we have stated this in the amended version of the manuscript (Page 14, lines 411-412)

Reviewer 2 Report

Comments and Suggestions for Authors

Review of the manuscript “Impact of sarcopenia in survival of patients with hepatocellular carcinoma treated with sorafenib”

written by Maurizio Biselli, Nicola Reggidori, Massimo Iavarone, Matteo Renzulli, Lorenzo Lani, Alessandro Granito, Fabio Piscaglia, Stefania Lorenzini, Eleonora Alimenti, Giulio Vara, Paolo Caraceni, Angelo Sangiovanni, Massimo Marignani, Elia Gigante, Nicolò Brandi, Annagiulia Gramenzi and Franco Trevisani.

I’ll write right away that I liked the manuscript. Dedicated to an important topic that is needed by society. I am writing not as a person in whose family there was a loss due to hepatocarcinoma, but as a researcher dealing with related issues. I have studied the manuscript quite carefully; in fact, I have no proposals. In this case, correction of the manuscript is simply necessary. 

1. The introduction of the manuscript is written extremely briefly. I suggest that the authors somewhat expand the introduction and highlight issues related to the prevalence of hepatocarcinoma, treatment of hepatocarcinoma, more thoroughly show the connection between sarcopenia and hepatocarcinoma, and clearly formulate the hypothesis. 

2. Discussion. We probably need to focus more on what follows from the relationship shown by the authors. What do the authors think? 

3. Minor flaws. Check the endings of sentences; in some places there are two dots instead of one dot. In some places there are inaccurate formulations. Can Kaplan-Meier rafts be colored? Is it possible to divide tables under Kaplan-Meier rafts with lines to make them easier to use?

Author Response

I’ll write right away that I liked the manuscript. Dedicated to an important topic that is needed by society. I am writing not as a person in whose family there was a loss due to hepatocarcinoma, but as a researcher dealing with related issues. I have studied the manuscript quite carefully; in fact, I have no proposals. In this case, correction of the manuscript is simply necessary.

Author Response: Thank you!

  1. The introduction of the manuscript is written extremely briefly. I suggest that the authors somewhat expand the introduction and highlight issues related to the prevalence of hepatocarcinoma, treatment of hepatocarcinoma, more thoroughly show the connection between sarcopenia and hepatocarcinoma, and clearly formulate the hypothesis.

    Author Response: we thank the reviewer for this comment. Accordingly, most part of the introduction has been rewritten according to your remarks and those of the reviewer 1 and a clearer hypothesis has been formulated  (page 2, lines 50-51, 54-60, 72-74, 87-91)

  2. We probably need to focus more on what follows from the relationship shown by the authors. What do the authors think?

    Author comment: we thank the reviewer for the suggestion. According to this, we wrote in the discussion a sentence more clearly against the use of Sorafenib in people with MELD > 9 and with sarcopenia (page 14, lines 391-392). On the other hand, we must recognize that our study has some limitations (retrospective design, relatively small population, absence of an untreated group) and therefore is "underpowered" to give a stronger indication (page 14, lines 405-408). 

  3. Minor flaws.
    a. Check the endings of sentences; in some places there are two dots instead of one dot.
    Author Response: done

    b.In some places there are inaccurate formulations
    Author Response: checked and improved

    c.Can Kaplan-Meier rafts be colored?
    Author Response: Yes, we'll collaborate with editorial staff if the paper will be accepted

    d. Is it possible to divide tables under Kaplan-Meier rafts with lines to make them easier to use?
    Author Response: Done

Reviewer 3 Report

Comments and Suggestions for Authors

I thank the editors for the opportunity to review the manuscript by Biselli et al. In their work, the authors present an analysis of the impact of sarcopenia on long-term outcomes for patients with HCC receiving a therapy with sorafenib. I can only congratulate the authors for their fine work, thought-out analysis and well-written manuscript, that contributes to the ongoing discussion on the impact of sarcopenia on cancer therapy outcomes. I have no further comments and recommend to accept the manuscript in its present form.

Author Response

Thank you so much! We greatly appreciate the reviewer’s comments.

Round 2

Reviewer 1 Report

Comments and Suggestions for Authors

All necessary corrections were done.